SciPost Physics

Submission

# The Loschmidt Index

Diego Liska[1*], Vladimir Gritsev[1,2],

**1** Institute for Theoretical Physics, Universiteit van Amsterdam, Science Park 904,
Postbus 94485, 1098 XH Amsterdam, The Netherlands
**2** Russian Quantum Center, Skolkovo, Moscow, Russia

*d.liska@uva.nl

## Abstract

We study the nodes of the wavefunction overlap between ground states of a parameter-dependent Hamiltonian. These nodes are topological, and we can use them to analyze in a unifying way both equilibrium and dynamical quantum phase transitions in multi-band systems. We define the Loschmidt index as the number of nodes in this overlap and discuss the relationship between this index and the wrapping number of a closed auxiliary hypersurface. This relationship allows us to compute this index systematically, using an integral representation of the wrapping number. We comment on the relationship between the Loschmidt index and other well-established topological numbers. As an example, we classify the equilibrium and dynamical quantum phase transitions of the XY model by counting the nodes in the wavefunction overlaps.

# 1 Introduction

Quantum phase transitions have been extensively studied using tools from quantum information theory and quantum geometry. The overlap between ground states evaluated at different parameters can be used to analyze the behaviour of systems undergoing quantum phase transitions [1–5]. The relevant quantities, like the Berry curvature and the fidelity susceptibility, are defined when these parameters are infinitesimally close to each other. The refinement of these two concepts is known nowadays as the Quantum Geometric Tensor (QGT) [6, 7]. The real part of the QGT defines a Riemann metric in the parameter space known as the quantum metric tensor or quantum information metric, while the imaginary part corresponds to the Berry curvature.

These geometric tensors have been used to create ground-state preparation protocols [8] and to derive bounds on the energy fluctuations over unit fidelity protocols [9]. Moreover, this approach to quantum mechanics can be experimentally tested in a number of different setups [10–17]. Yet, in recent years, experiments with atomic quantum gases [18–21] have led to the study of the wavefunctions overlaps when the two parameters are "far away". With these experiments we can test out of equilibrium quantum many-body systems. These systems cannot be captured within the typical thermodynamic description, and this has led to the development of new theoretical techniques and the discovery of phenomena that were not accessible within the framework of quantum statistical physics. One of these findings is that of dynamical quantum phase transitions (DQPT) [22, 23]. These are phase transitions driven by the progression of time itself, and not by external parameters such as temperature, pressure or magnetic fields.

DQPTs are defined using the squared overlap between a fixed and a time-evolved state

$$\mathcal{L}(t) = |\langle \Omega(0)|\Omega(t)\rangle|^2; \tag{1}$$

this overlap is a particular form of what is commonly known as the Loschmidt echo. Because of its resemblance to partition functions [23], the Loschmidt amplitude $\langle \Omega(0)|\Omega(t)\rangle$ is often put into the functional form $\mathcal{L} \sim \exp[-Ng(t)]$, where $N$ corresponds to the number of degrees of freedom of the system. There are a few exceptions to this behaviour that involve superextensive energy fluctuations in the system , see e.g. [24], but we restrict ourselves to the conventional exponential dependence of the degrees of freedom. This dependence allows us to define the rate function

$$R(t) = -\frac{1}{N} \log \left(|\langle \Omega(0)|\Omega(t)\rangle|^2\right), \tag{2}$$

which has a well-defined thermodynamic limit [23]. DQPTs correspond to nonanalytic points in this rate function. Crucially, the critical times that mark DQPT sit at a finite distance from the initial value $t = 0$.

There are two types of DQPTs: the so-called topological or symmetry-protected and accidental DQPTs [23, 25, 27]. Topological DQPTs happen for quenches between Hamiltonians with different topological properties. These phase transitions are robust in the sense that they do not depend on the details of the system. On the other hand, accidental DQPTs can be observed in quenches within the same topological phase. These DQPTs are not topologically protected and they, in general, depend on the details of the Hamiltonians [28–31].

For topological, multiband systems, it was recently understood [22, 25, 26] that some information about dynamical and equilibrium quantum phase transitions is encoded in the nodes of the wave function overlaps. In this paper, we propose a systematic way to count these nodes and show how we can use them to study equilibrium and dynamical phase transitions in a unifying way.

There are a few results that are particularly relevant to our work. The first result states that the overlap of two Bloch bands $|\psi_k\rangle$ and $|\phi_k\rangle$ with different topological indices must have at least one node in the Brillouin zone, i.e. there must be a vanishing overlap $\langle\psi_k|\phi_k\rangle$ for at least one momentum point $k$ [25, 26]. Furthermore, these nodes are connected to other topological numbers, for instance, in [25], the authors showed that in two-dimensional lattices, if $C_\psi$ and $C_\phi$ are the Chern numbers of the Bloch bands $|\psi_k\rangle$ and $|\phi_k\rangle$ respectively, then the wavefunction overlap must have at least $|C_\psi - C_\phi|$ nodes in the Brillouin zone. For one-dimensional systems, if the Berry phases of the Bloch bands are different, then there must be at least one node in the overlap.

The second statement applies to quench protocols. In a quench protocol, a state is prepared in the ground state $|\Omega(\lambda_o)\rangle$ of a parameter-dependent Hamiltonian $H(\lambda_o)$. Then, by a sudden change in a physical parameter, we evolve the system with a new Hamiltonian $H(\lambda_f)$. Thus, our time-evolved state is $\exp[-iH(\lambda_f)t]|\Omega(\lambda_o)\rangle$. The second result is the observation that, at the critical time $t_c$ of a DQPT, the squared overlap

$$\mathcal{L}_k(t) = |\langle\Omega_k(\lambda_o)|\exp[-iH_k(\lambda_f)t]|\Omega_k(\lambda_o)\rangle|^2 \tag{3}$$

must vanish for at least one momentum $k$ [22, 23]. For one-dimensional systems this is easily observed from the structure of $R(t)$, in the thermodynamic limit:

$$R(t) = -\frac{1}{2\pi}\int_{k\in\text{BZ}}\log\big[\mathcal{L}_k(t)\big]dk. \tag{4}$$

Hence, nonanalytic points in $R(t)$ correspond to nodes in $\mathcal{L}_k(t)$. These results emphasize the connection between topological nodes and phase transitions.

This paper wants to develop these ideas further and identify and classify quantum phase transitions only by node counting. We will explain how to systematically find these nodes for both dynamical and equilibrium quantum phase transitions in the next two sections. We will explain their connection to half-integer valued wrapping numbers. We will also discuss the differences and similarities between these nodes and other well-established topological numbers. Then, in section 4, we will use these concepts to classify the different quantum phase transitions of the XY model.

## 2 Nodes and wrapping numbers

Let us first show the relation between the nodes of a smooth periodic function $f(k)$ and the winding number of the closed planar curve $\alpha(k) = \big(x(k) = f'(k), y(k) = f(k)\big)$ in the $xy$-plane. We assume that the function $f(k)$ crosses the $k$-axis with a slope, i.e. $f'(k_o) \neq 0$ for $f(k_o) = 0$. Then, since the function has a period $L$, $f(k)$ must have at least two zeros. This means that the planar curve $\alpha(k)$ has a winding number of at least one. Fig. (1,a) makes this relationship clear. We define the winding number $W$ as the difference $\big[\theta(L) - \theta(0)\big]/(2\pi)$, where $\theta(k)$ is the angle the vector $\alpha(k)$ makes with respect to the $k$-axis. More precisely,

$$W[\alpha(k)] = \frac{1}{2\pi}\int_k d\theta(k) = \frac{1}{2\pi}\int_0^L \theta'(k)dk. \tag{5}$$

Since $\tan\theta(k) = x(k)/y(k)$ we can write

$$W[\alpha(k)] = \frac{1}{2\pi}\int_0^L \frac{f'(k)^2 - f(k)f''(k)}{f(k)^2 + f'(k)^2}dk. \tag{6}$$

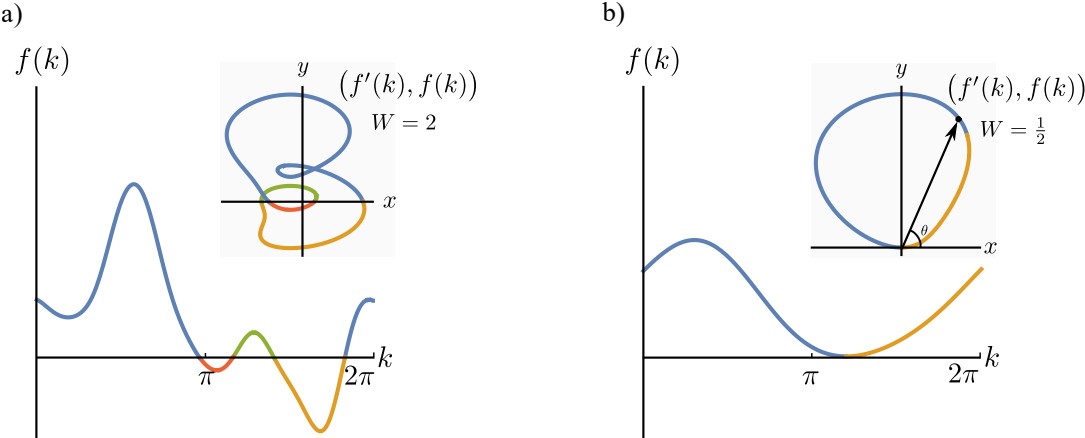

Figure 1: Relationship between the nodes of $f(k)$ and the winding number $W$ of the planar curve $\alpha(k) = \big(f'(k), f(k)\big)$.

If the function $f(k)$ touches the $k$-axis at $k_o$ with an even multiplicity, like in Fig.(1,b), then we have that $\alpha(k_o) = (0,0)$. Since the loop lies on the upper $xy$-plane, $\theta(L) = \pi$ instead of $2\pi$ and we find that $W = 1/2$. Therefore, we find that the number of nodes $n$ of the function $f(k)$ in its domain $k \in (0, L]$ is twice the winding number of $\alpha(k)$,

$$n[f(k)] = 2|W[\alpha(k)]|. \tag{7}$$

Note that the winding number of $\alpha(k)$ cannot be negative because when the derivative of $f$ is positive $(x > 0)$ the function is increasing and when the derivative is negative $(x < 0)$ the function is decreasing. As a result, the angle $\theta(k)$ is always an increasing function. The positivity of $W$ only holds for functions of one variable.

Since we are interested in squared overlaps, our nodes will always have an even multiplicity. This restriction will simplify our discussion in higher dimensions. For a more general discussion that considers non-periodic functions and node multiplicity, we recommend the Ref. [32]. Curiously, these ideas have led to the definition of non-integer valued winding numbers, and a generalized residue theorem [33].

To count nodes in two dimensions, we consider a smooth function $f(k^1, k^2)$ defined over a torus $(k^1, k^2) \in T^2 = (0, 2\pi] \times (0, 2\pi]$. We have rescaled our variables so that both have a period of $2\pi$. For convenience we denote $f(k^1, k^2)$ as $f(k^i)$. Now, instead of a loop in $\mathbb{R}^2$ we now have the surface $\alpha(k^i) = \big(x(k^i) = \partial_1 f(k^i), y(k^i) = \partial_2 f(k^i), z(k^i) = f(k^i)\big)$ embedded in $\mathbb{R}^3$. The intuition is the same as with the one-dimensional case, namely if the surface $f(k_o^i)$ crosses the $k^1 k^2$-plane with non-zero partial derivatives then $\alpha(k_o^i)$ must lie inside the $xy$-plane. This is what we see in Fig.(2,a). However, the situation in two dimensions is more complicated.

A surface $f(k^i)$ can intersect the plane at an isolated point $k_o^i$ or at a set of continuous points. The case of an isolated point $k_o^i$ is the easiest. We see in Fig.(2,d) that the normalized vector $\alpha(k^i)/|\alpha(k^i)|$ covers half of the unit 2-sphere. We can formalize this intuition by defining the wrapping number

$$W[\alpha(k^i)] = \frac{1}{\Omega_2} \int_{T^2} d\Omega_2(k^i), \tag{8}$$

where $\Omega_2 = 4\pi$ is the volume of the 2-sphere and $d\Omega_2(k^i) = \sin\big[\theta(k^i)\big] d\theta(k^i) d\phi(k^i)$ is the solid angle element of $\mathbb{R}^3$ given by the usual coordinates: $x = \sin\theta\cos\phi$, $y = \sin\theta\sin\phi$

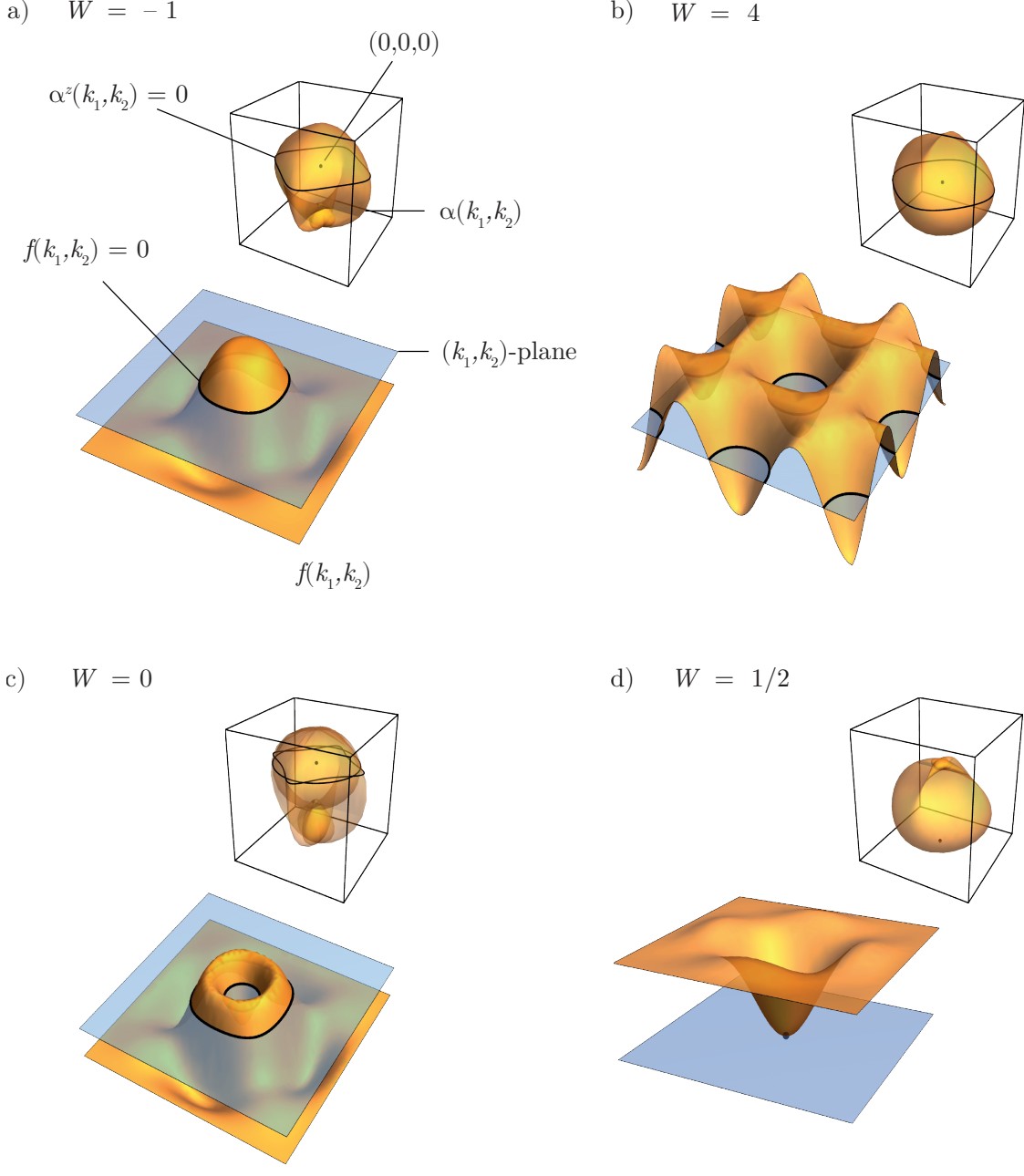

a)   $W = -1$

$(0,0,0)$

$\alpha^z(k_1, k_2) = 0$

$\alpha(k_1, k_2)$

$f(k_1, k_2) = 0$

$(k_1, k_2)$-plane

$f(k_1, k_2)$

b)   $W = 4$

c)   $W = 0$

d)   $W = 1/2$

Figure 2: Smooth periodic functions $f(k^1, k^2)$ and their corresponding closed surface $\alpha(k^1, k^2)$ and winding number $W$. Note that the winding number can be positive or negative. Functions with isolated nodes such as d) are related to phase transitions.

and $z = \cos\theta$. Changing from spherical to rectangular coordinates we find that

$$W[\alpha(k)] = \frac{1}{4\pi} \int_0^{2\pi} \int_0^{2\pi} \frac{\alpha \cdot (\partial_1\alpha \times \partial_2\alpha)}{|\alpha|^3} \, dk^1 dk^2. \tag{9}$$

This wrapping number has a sign depending on the function $f$; we see that $W[f] = -W[-f]$. So in two dimensions, the wrapping number of the surface $\alpha(k^i)$ can be negative. For functions with isolated nodes, e.g. non-negative or non-positive $f$, the absolute value of twice the winding number counts the number of nodes. Thus, we recover Eq.(7). In a more general setting, the total winding number can add up to zero, even if $f$ crosses the $k^1 k^2$-plane multiple times, see Fig.(2,c).

In this paper we will only encounter isolated nodes. Still, we would like to mention a few of the properties that we observed when there is a continuous set of points $K_o$. Intuitively, $W[\alpha]$ counts the number of loops inside $K_o$, see Figs.(2, a, b,c). Note that loops can add up or subtract each other, as it is the case of (2,b,c). If $K_o$ is not made of non-intersecting loops then, in general, the winding number $W[\alpha]$ can be ill defined. We chose the auxiliary surface $\alpha(k^i)$ because it was a natural generalization of the one-dimensional case to count isolated nodes. Moreover, it is easy to generalize to an arbitrary number of dimensions. However, if we instead want to count other features of $f$, then a different auxiliary surface could be more efficient depending on the situation.

For finite set of variables $(k^1, ..., k^n) \in T^n$ we can define the hypersurface $\alpha(k^i) = \left(\partial_1 f(k^i), ..., \partial_n f(k^i), f(k^i)\right)$. The wrapping number of this hypersurface is given by

$$W[\alpha(k^i)] = \frac{1}{\Omega_n} \int_{T^n} \alpha^* d\Omega_n, \tag{10}$$

where $d\Omega_n$ is the $n$-sphere volume element in $\mathbb{R}^{n+1}$ and $\Omega_n$ its volume, $\alpha^*$ is the pullback of $\alpha : T^n \to \mathbb{R}^{n+1}$. For smooth, non-negative functions $f(k^i)$ the number of nodes $n[f(k^i)]$ in the domain $T^n$ is twice the absolute value of the wrapping number $W[\alpha(k^i)]$.

As a final remark, we would like to point out the difference between Eq. (9) and the conventional Chern number. Although our expressions look remarkably similar to that of the Chern number, these are two different concepts. The Chern number of two-dimensional topological insulators is a topological invariant defined in a $U(n)$-fibre bundle. For two-dimensional band insulators, the Chern number of a Bloch vector $|\phi(t)\rangle$ also happens to be a wrapping number, but it is the wrapping number of a different map that takes pure states from $\mathbb{C}^2$ to the Bloch sphere. Nevertheless, interesting relationships exist between the nodes of the wavefunction overlaps and the topological numbers of the different Bloch bands. We have mentioned a few examples where lower bounds for these nodes can be computed using Chern numbers or Berry phases.

For DQPTs, one can also define a Chern number to characterize the phase transition. The expression for this number is remarkably similar to that of Eq. (9), but again, the maps to the sphere are different, so the two winding numbers have different interpretations. We will give more details about these two numbers in the next section.

## 3 The Loschmidt Index

Let us consider a Hamiltonian $H(\lambda) = \sum_k H_k(\lambda)$. Here, $\lambda^\mu$ is a vector of continuous parameters and $k \in [0, 2\pi)$. Because of the different momentum sectors $k$ are decoupled, the Hamiltonian can be diagonalized for each momentum sector $k$ separately. Hence, the ground state of this Hamiltonian exhibits the factorization property

$$|\Omega(\lambda)\rangle = \prod_{k \in \text{BZ}} |\Omega_k(\lambda)\rangle, \tag{11}$$

where $|\Omega_k(\lambda)\rangle \in \mathbb{C}^n$. The function $f(k) = |\langle\Omega_k(\lambda_1)|\Omega_k(\lambda_2)\rangle|^2$ must have at least one node whenever $\lambda_1$ and $\lambda_2$ correspond to different topological numbers. We can count the nodes of $f$ using the integral in Eq. (6), and the resulting integer $n[f(k)] = 2W[\alpha(k)]$ must be larger than one if there is if $\lambda_1$ and $\lambda_2$ correspond to different phases of matter. We define the *Loschmidt index* $n(\lambda_1, \lambda_2)$ between $|\Omega_k(\lambda_1)\rangle$ and $|\Omega_k(\lambda_2)\rangle$ to be the number of nodes in the wavefunction overlap i.e., $n(\lambda_1, \lambda_2) = n[f(k)]$. We note that the form of the wave function in (11) in a form as a product state is not very crucial; however in this form it is easy to define the thermodynamic limit as e.g. in Eq. (2) and to deal with a continuous label for the $f(k)$.

Dynamical quantum phase transitions behave differently. A DQPT happens when the overlap $\mathcal{L}_k(t)$ in Eq.(3) vanishes for at least one momentum $k_c$ at the critical time $t_c$, i.e. $\mathcal{L}_{k_c}(t_c) = 0$. For all other times $t$ and momenta $k$, $\mathcal{L}_k(t) > 0$. Thus, for these phase transitions, we have to count the number of nodes of the function $f(t, k) = R_k(t)$ of two variables $(t, k)$. For a general function $f$, this is problematic because $t$ is not necessarily a periodic variable. Still, in many cases, $t$ is a periodic variable for each momentum sector $k$, so we can use Eq.(9) to compute the number of nodes $m(\lambda_o, \lambda_f) = m[R_k(t)]$. Again, $m > 0$ if there is a DQPT in the quench protocol between the ground state of $H(\lambda_o)$ and the Hamiltonian $H(\lambda_f)$. Note that $m$ is still a Loschmidt index, the only difference is that now we have to integrate over two continuous variables $t$ and $k$. We will use $n$ for equilibrium phase transitions and $m$ for DQPTs.

There are at least two other notions of topological invariants for DQPTs in the literature [34–36]. The topological invariant described by Budich and Heyl [36] is time-dependent. This invariant is defined using the winding number of the Pancharatnam geometric phase, and it changes its integer value at each critical time $t_c$. The invariant introduced by Yang, Li and Chen [34] is a topological invariant for two-band models. The authors define several Chern numbers using the map $|\Omega_k(t)\rangle \rightarrow \rho_k(t) = |\Omega_k(t)\rangle\langle\Omega_k(t)|$ from $\mathbb{C}^2$ to the Bloch sphere and taking into account the fixed points of this map. These Chern numbers $C^m$ count how many times each map, defined from fixed point to fixed point, wraps the Bloch sphere, and only depend on the initial and final parameters $C^m = C^m(\lambda_o, \lambda_f)$. Unlike the number of nodes, these Chern numbers are only nontrivial when the DQPT is topologically protected [34]. The topological invariant of Yang, Li and Chen was recently generalized in [35] to quantum quenches of finite duration.

From the perspective of the present work one advantage of studying topological nodes is that there is no need to first map the solutions to the Bloch sphere –we only need the squared overlaps. This also allows us to generalize these results to multi-band topological systems.

# 4  The XY model

Now, let us apply these concepts to study the phase transitions of the XY model, a "drosophila" of quantum phase transitions in many-body physics. We start with the Hamiltonian

$$H = -\sum_{j=1}^{N} J_x\sigma_j^x\sigma_{j+1}^x + J_y\sigma_j^y\sigma_{j+1}^y + h\sigma_j^z \tag{12}$$

where $\sigma_j^\alpha$ are the Pauli matrices of the $j$-th spin site. To fix our energy scale we work with the variables:

$$J_x = J\left(\frac{1+\gamma}{2}\right) \text{ and } J_y = J\left(\frac{1-\gamma}{2}\right), \tag{13}$$

and set $J = 1$. This model can be solved using a Jordan-Wigner and a Fourier transformation. We will not solve this model here, for details see e.g. [37]. The two mappings turn our model into a fermionic system

$$H = \sum_k \vec{c}_k^\dagger H_k \vec{c}_k \quad \text{with} \quad \vec{c}_k = \begin{pmatrix} c_k \\ c_{-k}^\dagger \end{pmatrix}, \quad \vec{c}_k^\dagger = \left( c_k^\dagger, c_{-k} \right) \quad \text{and}$$

$$H_k = - \begin{pmatrix} \cos k - h & -i\gamma \sin k \\ i\gamma \sin k & h - \cos k \end{pmatrix}, \tag{14}$$

here the creation and annihilation operators $c$ and $c^\dagger$ obey the usual commutation relations: $\{c_k, c_{k'}^\dagger\} = \delta_{kk'}$ and zero for the other anticommutators. The mapping gives a unique ground state throughout the entire phase diagram

$$|\Omega(h, \gamma)\rangle = \prod_{k>0} |\Omega(h, \gamma)\rangle = \prod_{k>0} \left[ \cos\left( \frac{\theta_k}{2} \right) - i \sin\left( \frac{\theta_k}{2} \right) c_{-k}^\dagger c_k^\dagger \right] |0\rangle, \tag{15}$$

where

$$E_k = \sqrt{(h - \cos k)^2 + \gamma^2 \sin^2 k} \quad \text{and} \quad \tan \theta_k = \frac{\gamma \sin k}{h - \cos k}. \tag{16}$$

Since different momentum modes do not interact, we only need to study the properties of each momentum sector $|\Omega_k(\lambda)\rangle$, for $\lambda^\mu = (h, \gamma)$. It is often useful to introduce the Bloch vector $|\Psi_k(\lambda)\rangle = \left[ \sin(\theta_k/2) + i \cos(\theta_k/2) c_{-k}^\dagger c_k^\dagger \right] |0\rangle$. Note that $|\Psi_k(\lambda)\rangle$ and $|\Omega_k(\lambda)\rangle$ correspond to the eigenvectors of $H_{\pm k}$ with eigenvalues $E_{\pm k}$ and $-E_{\pm k}$ respectively.

## 4.1 Equilibrium quantum phase transitions

We start by studying equilibrium quantum phase transitions. The overlap between two ground states in each momentum sector is

$$f(\lambda_1, \lambda_2, k) = |\langle \Omega_k(\lambda_1) | \Omega_k(\lambda_2) \rangle|^2 = \cos^2 \left[ \frac{\Delta\theta_k}{2} \right]. \tag{17}$$

Here, $\Delta\theta_k = \theta_k(\lambda_2) - \theta_k(\lambda_1)$, and we have made explicit that $f(k) = f(\lambda_1, \lambda_2, k)$. Given $\lambda_1$ and $\lambda_2$, we can numerically compute the index $n(\lambda_1, \lambda_2)$ of the overlap.

In Fig.(3,a) and Fig.(3,b) we show two plots where $\lambda_1$ is fixed (black dot) and the value of $n(\lambda_1, \lambda)$ is computed for the entire phase diagram. We see that $n(\lambda_1, \lambda)$ divides the phase diagram into four or five different regions depending on the fixed value $\lambda_1$. The lines that remain fixed for all $\lambda_1$ correspond to the critical lines of the XY model. They separate the ferromagnetic phases from themselves and from the paramagnetic phase.

Note that the lines that do not correspond to different quantum phases of matter are not universal, meaning that they depend on the value of $\lambda_1$. These lines divide parameter space into new regions that behave similar to a situation with the accidental DQPT. To see this, let us pick a curve $\lambda(t) = (h(t), \gamma(t))$ that crosses all the regions of our phase diagram, e.g. the curve in Fig.(3,b). Instead of a quantum quench, we choose to evolve our system adiabatically. We can do this usig different Hamiltonians. DQPTs have been considered recently by many authors [35, 38–42] in the context of adiabatic, finite duration protocols or slow quenches. For example, a counteradiabatic protocol with a Hamiltonian $H_{\text{CD}}(t)$ (CD=counteradiabatic) such that

$$|\Omega(t)\rangle = \exp\left[ -it H_{\text{CD}}(t) \right] \left| \Omega\big(h(0), \gamma(0)\big) \right\rangle = |\Omega(h(t), \gamma(t))\rangle. \tag{18}$$

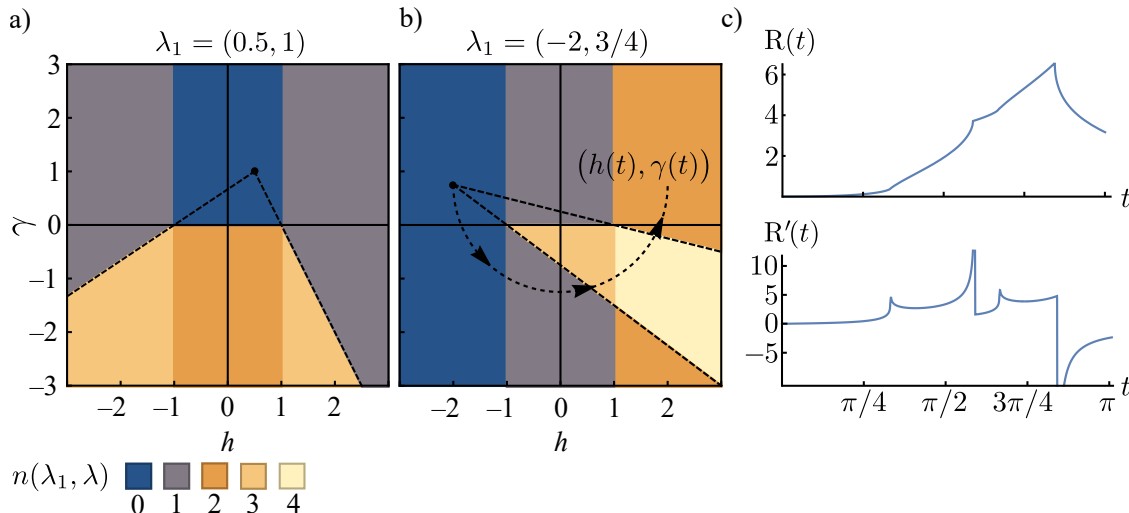

Figure 3: a) and b) The Loschmidt index $n(\lambda_1, \lambda)$ throughout the phase diagram of the overlap $f(k) = |\langle \Omega_k(\lambda_1)|\Omega_k(\lambda)\rangle|^2$ for a fixed value $\lambda_1$ (black point). Note how we can use this index to identify equilibrium quantum phase transitions. c) Plot of the rate function $R(t) = -\log\left[|\langle\Omega(0)|\Omega(h(t),\gamma(t))\rangle|^2\right]/N$ and its derivative. There is a nonanalytic point each time the curve crosses a critical line.

will force our system to remain in the ground-state manifold [43–46]. The time $t$ in this evolution depends on our choice of Hamiltonian; we should think about it as a variable parametrizing the curve inside the ground-state manifold. As our system evolves, we can measure the rate function $R(t) = -\log\left[|\langle\Omega(0)|\Omega(h(t),\gamma(t))\rangle|^2\right]/N$ of the Loschmidt echo. As we see in Fig.(3,c), each time we cross a critical line, the rate function exhibits nonanalytic behavior either in $R(t)$ or in its derivative $R'(t)$. As with accidental DQPT, these phase transitions are not universal and depend on the details of the protocol.

## 4.2 Dynamical quantum phase transitions

Now, we would like to study the nodes of DQPTs in the XY model. For simplicity, we will limit our discussion to quantum quench protocols. Due to the factorization property of our ground state, we have that

$$
\begin{aligned}
\langle\Omega_k(\lambda_o)|\, &e^{-it\left(H_k(\lambda_f)+H_{-k}(\lambda_f)\right)}\,|\Omega_k(\lambda_o)\rangle \\
&= \langle\Omega_k(\lambda_o)|\left(e^{2iE_k(\lambda_f)t}a_1\,|\Omega_k(\lambda_f)\rangle + e^{-2iE_k(\lambda_f)t}a_2\,|\Psi_k(\lambda_f)\rangle\right) \\
&= e^{2iE_k(\lambda_f)t}|a_1|^2 + e^{-2iE_k(\lambda_f)t}(1-|a_1|^2)
\end{aligned}
\tag{19}
$$

Here, $a_1 = \langle\Omega_k(\lambda_f)|\Omega_k(\lambda_o)\rangle = \cos(\Delta\theta_k/2)$ and $a_2 = \langle\Psi_k(\lambda_f)|\Omega_k(\lambda_o)\rangle = \sin(\Delta\theta_k/2)$, where $\Delta\theta_k = \theta_k(\lambda_f) - \theta_k(\lambda_o)$. Thus, we see that the overlap is periodic in the time and momentum coordinates. For simplicity, we rescale the time parameter $\tilde{t} = 2E_k t$. To classify the different DQPTs we investigate on the nodes of the function

$$
f\left(\lambda_o, \lambda_f, \tilde{t}, k\right) = \mathcal{L}_k(\tilde{t}) = \cos^2\left(\tilde{t}\right) + \cos^2(\Delta\theta_k)\sin^2\left(\tilde{t}\right),
\tag{20}
$$

Again, we have made explicit the dependence on the parameters $\lambda_o$ of the initial ground state and the parameter $\lambda_f$ of the final Hamiltonian. If we fix the value of $\lambda_o$ we can compute $m(\lambda_o, \lambda)$ for the phase diagram of this model. Fig.(4) shows the results of numerically

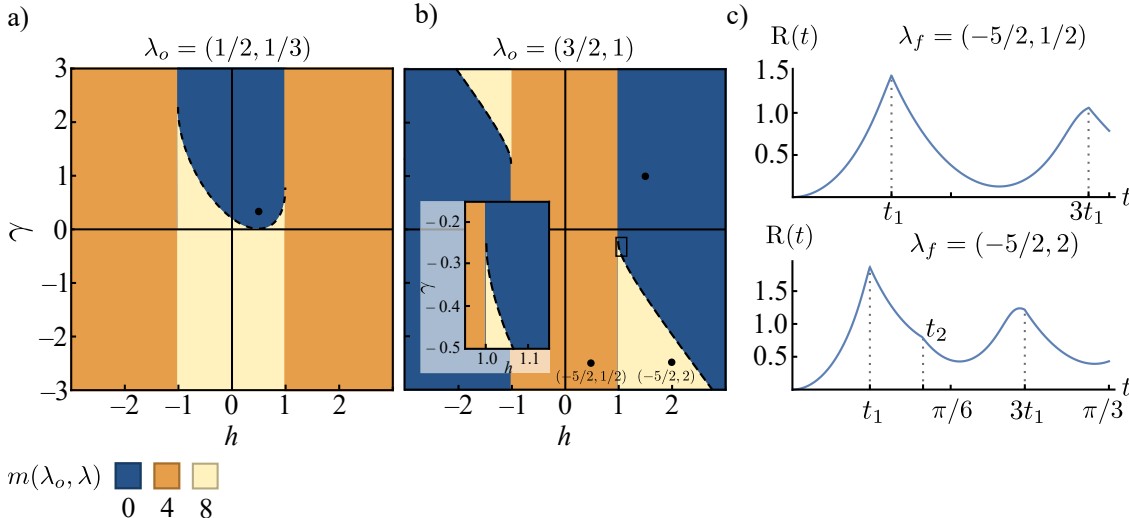

Figure 4: a) and b) The Loschmidt index $m(\lambda_o, \lambda)$ throughout the phase diagram of the squared overlap $f(t, k) = R_k(t)$ for a fixed value of $\lambda_o$ (black point). This index is sensitive to topologically protected and accidental DQPTs. c) Two plots of the rate function $R(t)$ for different values of $\lambda_f$ but the same value of $\lambda_o$. Note that $m(\lambda_o, \lambda_f)/4$ corresponds to the number of critical times $t_i$ in each plot.

integrating the Loschmidt index. We find the two types of DQPTs. We see that accidental DQPTs depend on the value of the initial state parameter $\lambda_o$ while topological protected phase transitions do not. As a self-consistency check, in appendix A, we show how to derive the exact equations of these accidental "critical" lines to check that they match the numerical results.

For this model and many others, see [23], $f(\lambda_o, \lambda_f, \tilde{t}, k) = 0$ only if $\cos(\tilde{t}) = 0$. So the critical times are fixed values $\tilde{t}_c = \pi/2,\ 3\pi/2 \mod 2\pi$. This means that $m(\lambda_o, \lambda_f)$ counts the sum of the number of nodes in the functions $f(\lambda_o, \lambda_f, \pi/2, k)$ and $f(\lambda_o, \lambda_f, 3\pi/2, k)$. For the XY model, this corresponds to four times the number of nodes $k_c$ in $\cos(\Delta\theta_k)$, $k \in [0, \pi]$. Here we used the symmetry $\Delta\theta_k = \Delta\theta_{-k}$, so we only have to consider half the domain of $k$. DQPTs appear in sets, labeled by critical momenta $k_c$: $t_c = (2l + 1)\pi/(4E_{k_c}(\lambda_f))$, for any integer $l$. So, $m/4$ counts how many of these sets a given quench protocol has. As we see in Fig.(4,c), a protocol with $m = 4$ has only one row of critical times $t_1(2l + 1)$, but a protocol with $m = 8$ has two rows $t_i(2l + 1)$ with $i = 1, 2$.

## 5 Conclusion

In this work, we studied equilibrium and dynamical quantum phase transitions using only the nodes of the ground-state wavefunction overlaps. We did this by introducing two integral representations that count these nodes. We showed the relationship between these nodes and the wrapping number of an auxiliary hypersurface, and defined the Loschmidt index as the number of nodes in the wavefunction overlaps. Our proposal is to use these numbers to study equilibrium and dynamical quantum phase transitions. We showed how this classification works for the XY model.

For equilibrium quantum phase transitions, we found that the corresponding overlaps can have 0, 1, 2, 3 or 4 nodes. The number of nodes splits the phase diagram into four or five regions. The lines that remain fixed for all values of the initial parameter $\lambda_1$ correspond

to the critical lines of the XY model. We showed that the non-universal regions resemble the behaviour of DQPT.

Finally, we analysed the DQPTs of the XY model when considering a quantum quench. This time, the number of nodes divides the phase diagram into three regions. We found that the Loschmidt index is four times the number of critical momenta $k_c \in [0, \pi]$. We saw that the number of nodes only depends on the initial ground-state parameter $\lambda_o$ and final parameter $\lambda_f$ of the Hamiltonian. As expected, lines that are independent of the details of the initial and final Hamiltonian separate regions that are topologically protected, while lines that depend on these details correspond to accidental DQPTs.

# 6   Acknowledgements

We would like to thank Anatoli Polkovnikov and Mikhail Pletyukhov for useful discussions. This work is part of the DeltaITP consortium, a program of the Netherlands Organization for Scientific Research (NWO) funded by the Dutch Ministry of Education, Culture and Science (OCW).

# A   Lines separating accidental DQPTs

In this appendix, we will derive the critical lines separating accidental DQPTs. Setting Eq.(20) to zero, we find that the nodes of the must satisfy $\tilde{t} = \pm\pi/2 \mod \pi$. So the critical times are fixed. Hence, a DQPT happens whenever there is a solution for the equation:

$$\cos \Delta\theta_k = 0. \tag{21}$$

To solve this equation, we define the vectors $v_{f/o} = (h_{f/o} - \cos k, \gamma_{f/o} \sin k)$. Such that

$$\cos \Delta\theta_k = \frac{v_o \cdot v_f}{|v_o||v_f|} = \frac{(h_o - \cos k)(h_f - \cos k) + \gamma_1 \gamma_2 \sin^2 k}{E_k(\lambda_o)^2 E_k(\lambda_f)^2} = 0. \tag{22}$$

We are interested in the critical lines of accidental DQPTs. If we imagine a set of protocols labeled by a continuous parameter $\sigma$, such that $\lambda_f(\sigma)$ crosses the critical line at $\sigma_c$, then we expect the node of the function $f_\sigma(k, \pm\pi/2)$ to appear smoothly, since for accidental DQPT, the system does not undergo a phase transition. That is, at the critical line there is a momentum $k$ such that

$$\cos \Delta\theta_k = 0 \quad \text{and} \quad \partial_k \cos \Delta\theta_k = 0. \tag{23}$$

The last equation implies that $\cos k = (h_o + h_f)/(2 - 2\gamma_o\gamma_f)$ or $\sin k = 0$. Plugging $k = 0, \pi$ into Eq. (22), we find that $h_f = \pm 1$. Therefore, accidental critical lines do not have $\sin k$ equal to zero. Substituting our expression for the cosine of $k$ into Eq.(22); we find that

$$4(\gamma_o\gamma_f - 1)(h_o h_f + \gamma_o\gamma_f) + (h_o + h_f)^2 = 0. \tag{24}$$

Thus, for initial parameters $(h_o, \gamma_o)$, accidental critical lines correspond to conic sections. These are the dotted curves in Fig.(4).

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
