# Peer review of "The Loschmidt Index"

_SciPost Physics_

## Round 1 · Referee Report · Anonymous (Referee 1) · 2021-3-30

Report

In this paper, the authors introduce a new topological metric, the Loschmidt index, which they show can be used to count zeros of a generic real-valued function f. They specifically focus on the case where f is the Loschmidt echo, whose zeros play a defining role in dynamical phase transitions. They demonstrate that this works both equilibrium and non-equilibrium phase transitions in the XY model.
This idea is new and relevant, and therefore I recommend publication in SciPost Physics. The paper is well-written and I don’t have major notes regarding the text. My main comment to the authors is that it would be useful to more clearly address how the method is expected to be useful. Since the topological invariant is constructed my mapping f to a d+1 dimensional function, one of whose parameters is f itself, it seems like one could simply count the zeros of f by putting the function on a grid and looking for zero crossings. This would be much easier than numerically calculating the higher-dimensional winding number integral. My question to the authors is what are the conditions under which the topological integral would be more useful than simple zero counting?
  • validity: high
  • significance: ok
  • originality: ok
  • clarity: high
  • formatting: excellent
  • grammar: excellent

Author:  Diego Liska  on 2021-04-22  [id 1377]

(in reply to Report 1 on 2021-03-30)

Dear Referee,

Thank you for your report. We have addressed your comment in the paper in the paragraph following Eq. (10). Our invariant was engineered to count these zeros, so a grid indeed suffices in one or two dimensions. In higher dimensions, this is more difficult to visualize. We are working with a positive function with a finite number of zeros, so we cannot look for zero crossings. To count these zeros, we have to find all local minima and check if they are zero or not. We have to implement this procedure numerically, and a straightforward way to do it is by doing the integral on our paper. One advantage of doing the integral is that we know it evaluates to an integer, so we do not have to compute it with really high precision. It is also easier to do the integral when evaluating entire regions of parameter space since our expressions depend explicitly on the variables of the system.

Kind Regards,

D. Liska
V. Gritsev

---

## Round 1 · Referee Report · Anonymous (Referee 2) · 2021-4-8

Report

In this work the authors study ground state properties of noninteracting topological systems and their implications on nonequilibrium real-time dynamics in the context of dynamical quantum phase transitions. More specifically, they investigate overlaps of ground state wave functions of a parameter-dependent Hamiltonian. They show that such overlaps can exhibit nodes which are topological and which can be used to classify both the system's equilibrium and nonequilibrium properties. In particular, they introduce a so-called Loschmidt index, which is the number of nodes in these wavefunction overlaps. Importantly, their analysis not only applies to the mostly studied two-band models but can be equally applied to multi-band systems. Finally, the authors apply their theory to a paradigmatic model system, the so-called one-dimensional XY model and they discuss thoroughly the connection to other previously introduced topological classifications in this context.

The manuscript is well-structured and well written and is accessible also to non-expert readers. The results are sound and the analysis thorough. The main topic covered in this work is also timely. Overall, I consider this a nice piece of work and I am therefore in favor of recommending it for publication in SciPost.

I could, however, imagine that the present manuscript could profit significantly from also applying their theory to a regime, which could particularly highlight the strength of the introduced Loschmidt index. In the presentation the authors already consider the aforementioned XY model, which, however, is relatively simple and whose properties have been already very well described by other topological quantum numbers introduced recently. In particular, the XY model realizes a two-band model. The challenging regime is the multi-band case and I would consider the present manuscript an even stronger contribution when their theory would have been applied to such a more difficult case.

While I wouldn't consider this point as a mandatory addition, I feel that the manuscript could profit significantly from extending the discussion to the multi-band case.
  • validity: top
  • significance: good
  • originality: high
  • clarity: high
  • formatting: excellent
  • grammar: excellent

Author:  Diego Liska  on 2021-04-22  [id 1376]

(in reply to Report 2 on 2021-04-08)

Dear Referee,

Thank you for your comments on our paper. Indeed, we think the next step is to show how this machinery works on a multiband system. We are currently working on this subject, but we believe the discussion is worth extending to another paper. We wanted to keep this paper short and accessible, so we decided to do the relatively known and straightforward example of the XY model.

Kind Regards,

D. Liska
V. Gritsev

---

## Editorial Decision

resubmitted